# LoCoOp: Few-Shot Out-of-Distribution Detection via Prompt Learning

**Atsuyuki Miyai[1]  Qing Yu[1,2]  Go Irie[3]  Kiyoharu Aizawa[1]**
[1]The University of Tokyo   [2]LY Corporation   [3]Tokyo University of Science
{miyai,yu,aizawa}@hal.t.u-tokyo.ac.jp  goirie@ieee.org

## Abstract

We present a novel vision-language prompt learning approach for few-shot out-of-distribution (OOD) detection. Few-shot OOD detection aims to detect OOD images from classes that are unseen during training using only a few labeled in-distribution (ID) images. While prompt learning methods such as CoOp [60] have shown effectiveness and efficiency in few-shot ID classification, they still face limitations in OOD detection due to the potential presence of ID-irrelevant information in text embeddings. To address this issue, we introduce a new approach called **Lo**cal regularized **Co**ntext **Op**timization (LoCoOp), which performs OOD regularization that utilizes the portions of CLIP local features as OOD features during training. CLIP's local features have a lot of ID-irrelevant nuisances (*e.g.*, backgrounds), and by learning to push them away from the ID class text embeddings, we can remove the nuisances in the ID class text embeddings and enhance the separation between ID and OOD. Experiments on the large-scale ImageNet OOD detection benchmarks demonstrate the superiority of our LoCoOp over zero-shot, fully supervised detection methods and prompt learning methods. Notably, even in a one-shot setting – just one label per class, LoCoOp outperforms existing zero-shot and fully supervised detection methods. The code is available via https://github.com/AtsuMiyai/LoCoOp.

## 1   Introduction

Detecting out-of-distribution (OOD) samples is crucial for deploying machine learning models in real-world scenarios, where new classes of samples can naturally arise and require caution. Common approaches for OOD detection use single-modal supervised learning [15, 17, 19, 28, 45, 29, 26, 50, 1]. These supervised methods achieve good results but have limitations *e.g.*, these methods require a lot of computational and annotation costs for training. These days, CLIP [37] achieves surprising performances for various downstream tasks, and the application of CLIP to OOD detection is beginning to attract increasing attention [11, 30, 33, 46].

Previous studies on CLIP-based OOD detection have focused on extreme settings. Some studies [12, 11, 30, 33] have explored zero-shot OOD detection methods that do not require any in-distribution (ID) training data, whereas other studies [46] have developed fully supervised methods that require the entire ID training data. These methods have comparable performances, but both approaches have their own limitations. On the one hand, the zero-shot methods [30, 33] do not require any training data, but they may encounter a domain gap with ID downstream data, which limits the performance of zero-shot methods. On the other hand, fully supervised methods [46] utilize the entire ID training data but may destroy the rich representations of CLIP by fine-tuning, which also limits the performance despite requiring enormous training costs. To overcome the limitations of fully supervised and zero-shot methods, it is essential to develop a few-shot OOD detection method that utilizes a few ID training images for OOD detection. By finding a balance between the zero-shot

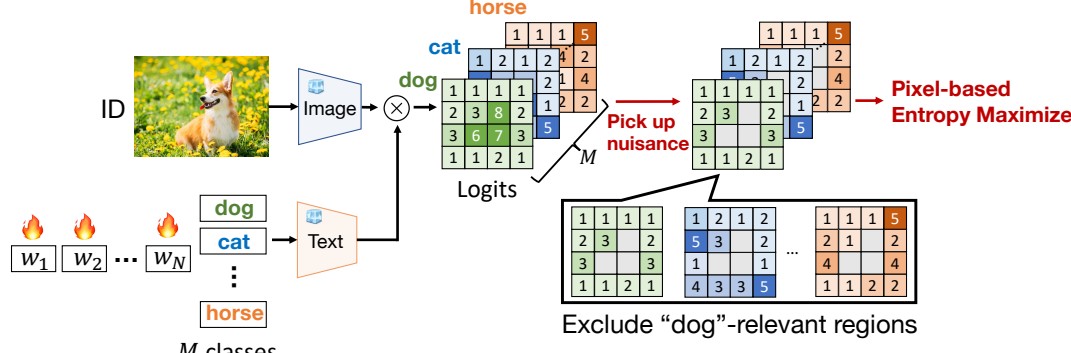

Figure 1: **Overview of the OOD regularization for our LoCoOp.** Our approach first extracts ID-irreverent nuisances (OOD) from CLIP's local features. Then, it performs entropy maximization on a per-pixel basis for the extracted OOD features. This OOD regularization allows OOD nuisances and all ID text embeddings to be dissimilar, thus preventing the inclusion of nuisances in ID text embeddings and improving test-time detection performance.

and fine-tuned methods, few-shot OOD detection has the potential to provide a more efficient and effective solution for OOD detection.

In this study, we tackle CLIP-based few-shot OOD detection, where we utilize only a few ID images for training CLIP. The most common approach to train CLIP with a few ID training data is prompt learning [60, 61], which trains the prompt's context words while keeping the entire pre-trained parameters fixed. CoOp [60] is the representative work of prompt learning, but we observe that CoOp does not still work well for OOD detection. While CoOp is designed to bring the given ID image and its corresponding class text embedding closer together, it also has the unintended effect of bringing the text embeddings closer to the backgrounds or ID-irrelevant objects in the ID image. Consequently, the text embeddings may contain information that is not relevant to the ID image, resulting in incorrectly high confidence scores for OOD images.

To apply prompt learning to OOD detection, we propose a simple yet effective approach called **Lo**cal regularized **Co**ntext **Op**timization (LoCoOp). An overview of our LoCoOp approach is shown in Fig. 1. We observe that the local features of CLIP contain many ID-irrelevant nuisances, such as backgrounds. LoCoOp treats such ID-irrelevant nuisances as OOD and learns to push them away from the ID class text embeddings, which removes unnecessary information from the text embeddings of ID classes and prevents the model from producing high ID confidence scores for the OOD features. To achieve this, LoCoOp first identifies the ID-irrelevant regions using the segmentation classification prediction results [59] for each local region. Next, it performs entropy maximization of the classification probabilities for the extracted regions, which ensures the features in ID-irrelevant regions are dissimilar to any ID text embedding.

Our proposed method, LoCoOp, offers two significant advantages. Firstly, it can perform OOD regularization at a minimal cost. The standard approach to OOD regularization is OOD exposure [16, 54, 52], which utilizes external OOD samples for training. However, it can be quite cumbersome to collect a lot of OOD images. In addition, it can be computationally expensive as the number of OOD images and training time are proportional. On the other hand, LoCoOp uses only local features of CLIP as OOD. Therefore, even when 100 OOD features are used per ID image, the training time is only about $1.4\times$ longer and GPU memory consumption is only about $1.1\times$ higher than those of the baseline method without OOD regularization (*i.e.*, CoOp). Secondly, LoCoOp is highly compatible with GL-MCM [33], a test-time OOD detection method that separates ID and OOD using both global and local features. Since LoCoOp learns local OOD features, it can accurately identify whether the given local region is ID or OOD. This enables LoCoOp to improve the OOD detection performance with GL-MCM, significantly outperforming other methods.

Through experiments on the large-scaled ImageNet OOD benchmarks [18], we observe that our LoCoOp significantly outperforms existing zero-shot, few-shot, and fully supervised OOD detection methods. Notably, even in a one-shot setting – just one label per class, LoCoOp outperforms existing

zero-shot and fully supervised detection methods, which is a remarkable finding. The contributions of our study are summarized as follows:

- We first tackle CLIP-based few-shot OOD detection, where we utilize only a few labeled ID images to train CLIP for OOD detection.
- We propose a novel prompt learning approach called LoCoOp, which leverages the portions of CLIP local features as OOD features for OOD regularization (see Fig. 1).
- LoCoOp brings substantial improvements over prior zero-shot, fully supervised, and prompt learning methods on the large-scale ImageNet OOD benchmarks. Notably, LoCoOp outperforms them with only one label per class (see Table 1).

## 2 Method

### 2.1 Problem statement

For OOD detection with pre-trained models [12, 30, 24], the ID classes refer to the classes used in the downstream classification task, which are different from those of the upstream pre-training. Accordingly, OOD classes are those that do not belong to any of the ID classes of the downstream task. An OOD detector can be viewed as a binary classifier that identifies whether the image is an ID image or an OOD image. Formally, we assume that we have an ID dataset $D^{\text{in}}$ of $(x^{\text{in}}, y^{\text{in}})$ pairs where $x^{\text{in}}$ denotes the input ID image, and $y^{\text{in}} \in Y^{\text{in}} := 1, ..., M$ denotes the ID class label. Let $D^{\text{out}}$ denote an OOD dataset of $(x^{\text{out}}, y^{\text{out}})$ pairs where $x^{\text{out}}$ denotes the input OOD image, and $y^{\text{out}} \in Y^{\text{out}} := M + 1, ..., M + O$ denotes the OOD class label. Then, there are no overlaps of classes between ID and OOD. Formally, $Y^{\text{out}} \cap Y^{\text{in}} = \emptyset$

We study the scenario where the model is fine-tuned only on the training set $D^{\text{in}}_{\text{train}}$ without any access to OOD data. The test set contains $D^{\text{in}}_{\text{test}}$ and $D^{\text{out}}_{\text{test}}$ for evaluating OOD performance. Unlike existing studies using a large number of ID training samples [46] or no ID training samples [30], we use only a few samples (1, 2, 4, 8, and 16 samples per class) for $D^{\text{in}}_{\text{train}}$.

### 2.2 Reviews of CoOp

CoOp [60] is a pioneering method that utilizes vision-language pre-trained knowledge, such as CLIP [37], for downstream open-world visual recognition. While CLIP uses manually designed prompt templates, CoOp sets a portion of context words in the template as continuous learnable parameters that can be learned from few-shot data. Thus, the classification weights can be represented by the distance between the learned prompt and the visual feature.

Given an ID image $x^{\text{in}}$, a global visual feature $f^{\text{in}} = f(x^{\text{in}})$ is obtained by the visual encoder $f$ of CLIP. Then, the textual prompt can be formulated as $t_m = \{\omega_1, \omega_2, ..., \omega_N, c_m\}$, where $c_m$ denotes the word embedding of the ID class name, $\omega = \{\omega_n|_{n=1}^N\}$ are learnable vectors where each vector has the same dimension as the original word embedding and $N$ denotes the length of context words. With prompt $t_m$ as the input, the text encoder $g$ outputs the textual feature as $g_m = g(t_m)$. The final prediction probability is computed by the matching score as follows:

$$p(y = m \mid x^{\text{in}}) = \frac{\exp\left(\text{sim}\left(f^{\text{in}}, g_m\right)/\tau\right)}{\sum_{m'=1}^{M} \exp\left(\text{sim}\left(f^{\text{in}}, g_{m'}\right)/\tau\right)}, \tag{1}$$

where $\text{sim}(\cdot, \cdot)$ denotes cosine similarity, and $\tau$ stands for the temperature of Softmax. The final loss $\mathcal{L}_{\text{coop}}$ is the cross-entropy loss with Eq. (1) and its ground truth label $y^{\text{in}}$.

### 2.3 Proposed approach

In this section, we present our LoCoOp approach for few-shot OOD detection. An overview of our proposed method is shown in Fig. 1. Our methods consist of two components. The first is to extract ID-irrelevant regions from CLIP local features, and the second is to perform OOD regularization training with the extracted regions. It performs these two operations for each iteration.

In the following, we will first provide a brief overview of how to obtain local features from CLIP. Then, we will describe our two key components: the method to extract object-irrelevant regions from ID images and the OOD regularization loss used during training.

**Preliminary.** To obtain local features of CLIP, we project the visual feature ${\boldsymbol{f}'}_i^{\text{in}}$ of each region $i$ from the feature map to the textual space [59, 42, 33] as follows:

$$\boldsymbol{f}_i^{\text{in}} = \text{Proj}_{v \to t}(v\left({\boldsymbol{f}'}_i^{\text{in}}\right)), \tag{2}$$

where $v$ denotes the value projection and $\text{Proj}_{v \to t}$ denotes the projection from the visual space into the textual space. These projections are inherent in CLIP and do not need to be trained. This local feature $\boldsymbol{f}_i^{\text{in}}$ has a rich local visual and textual alignment [59].

**Extraction of ID-irrelevant regions.** We need to select the region indices of ID-irrelevant regions from a set of all region indices $I = \{0, 1, 2, ..., H \times W - 1\}$, where $H$ and $W$ denote the height and width of the feature map. We handle this in a very simple and intuitive way. Similar to the segmentation task [59], we can obtain the classification prediction probabilities for each region $i$ during training by computing the similarity between the image features $\boldsymbol{f}_i^{\text{in}}$ of each region $i$ and the text features of the ID classes. The formulation is as follows:

$$p_i(y = m \mid \boldsymbol{x}^{\text{in}}) = \frac{\exp\left(\text{sim}\left(\boldsymbol{f}_i^{\text{in}}, \boldsymbol{g}_m\right)/\tau\right)}{\sum_{m'=1}^{M} \exp\left(\text{sim}\left(\boldsymbol{f}_i^{\text{in}}, \boldsymbol{g}_{m'}\right)/\tau\right)}. \tag{3}$$

When a region $i$ of $\boldsymbol{x}^{\text{in}}$ corresponds to a part of an ID object, the ground truth class $y^{\text{in}}$ should be among the top-$K$ predictions. In contrast, if a region $i$ is ID-irrelevant, such as the background, $y^{\text{in}}$ should not appear among the top-$K$ predictions because of the lack of semantic relationship between region $i$ and the ground truth label $y^{\text{in}}$. Based on this observation, we identify regions that do not include their ground truth class in the top-$K$ predicted classes as ID-irrelevant regions $J$. The formulation is as follows:

$$J = \{i \in I : \text{rank}(p_i(y = y^{\text{in}}|\boldsymbol{x}^{\text{in}})) > K\}, \tag{4}$$

where $\text{rank}(p_i(y = y^{\text{in}}|\boldsymbol{x}^{\text{in}}))$ denotes the rank of the GT class $y^{\text{in}}$ among all ID classes. Note that a set of regions $J$ of an image $\boldsymbol{x}^{\text{in}}$ is updated during training since $p_i(y = m \mid \boldsymbol{x}^{\text{in}})$ is updated.

This method may appear to be a parameter-dependent method since it depends on the hyperparameter $K$. However, we argue that the optimal $K$ is not difficult to search for. This is because, when applying OOD detection in the real world, it is assumed that the users understand what ID data is. For example, when ID is ImageNet-1K [5], users know the number of fine-grained classes or semantic relationships between classes in advance. Using prior knowledge, such as the number of fine-grained classes, helps determine the value of $K$ (up to top-$K$, the prediction may be wrong).

**OOD regularization.** With $J$ (a set of region indices of ID-irrelevant regions), we perform OOD regularization, which removes unnecessary information from the text feature. The OOD features $\boldsymbol{f}_j^{\text{in}}$ of each OOD region $j \in J$ should be dissimilar to any ID text embedding. Therefore, we use entropy maximization for regularization, which is used for detecting unknown samples during training [40, 55]. Entropy maximization makes the entropy of $p_j(y|\boldsymbol{x}^{\text{in}})$ larger and enables the OOD image features $\boldsymbol{f}_j^{\text{in}}$ to be dissimilar to any ID text embedding. The loss function for this regularization is as follows:

$$\mathcal{L}_{\text{ood}} = -H(p_j), \tag{5}$$

where $H(\cdot)$ is the entropy function and $p_j$ denotes the classification prediction probabilities for region $j \in J$.

**Final Objective.** The final objective is as follows:

$$\mathcal{L} = \mathcal{L}_{\text{coop}} + \lambda \mathcal{L}_{\text{ood}}, \tag{6}$$

where $\mathcal{L}_{\text{coop}}$ is the loss for CoOp [60] and $\lambda$ is a hyperparameter. Here, $\mathcal{L}_{\text{coop}}$ is applied for the entire image.

## 2.4 Test-time OOD detection

In testing, we use MCM score [30] and GL-MCM score [33]. With these functions, we classify test-time images into $\boldsymbol{x}^{\text{in}}$ and $\boldsymbol{x}^{\text{out}}$. The details are as follows:

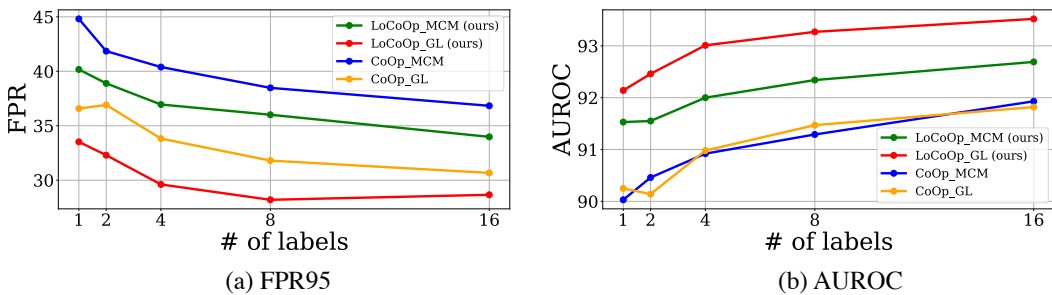

(a) FPR95          (b) AUROC

Figure 2: **Few-shot OOD detection results** with different numbers of ID labeled samples. We report average FPR95 and AUROC scores on four OOD datasets in Table 1. The lower value is better for FPR95, and the larger value is better for AUROC. We find that in all settings, our proposed LoCoOp with GL-MCM (red one) outperforms CoOp by a large margin.

**MCM score.** The MCM score [30] utilizes the softmax score of global image features and text features. Formally, $S_{\text{MCM}} = \max_m \frac{\exp(\text{sim}(\boldsymbol{f}, \boldsymbol{g}_m)/\tau)}{\sum_{m'=1}^{M} \exp(\text{sim}(\boldsymbol{f}, \boldsymbol{g}_{m'})/\tau)}$, where $\tau = 1$. The rationale is that, for ID data, it will be matched to one of the prototype vectors with a high score and vice versa.

**GL-MCM score.** The GL-MCM score [33] utilizes the softmax score of both global and local image features and text features. The score function is $S_{\text{GL-MCM}} = S_{\text{MCM}} + \max_{m,i} \frac{\exp(\text{sim}(\boldsymbol{f}_i, \boldsymbol{g}_m)/\tau)}{\sum_{m'=1}^{M} \exp(\text{sim}(\boldsymbol{f}_i, \boldsymbol{g}_{m'})/\tau)}$. When OOD objects appear in the given ID image, the matching score with global features might be incorrectly low. Utilizing the matching score with local features can compensate for the low global matching score and produce correct ID confidence.

## 3 Experiment

### 3.1 Experimental Detail

**Datasets.** We use the ImageNet-1K dataset [5] as the ID data. Therefore, $M$ is set to 1,000. For OOD datasets, we adopt the same ones as in [18], including subsets of iNaturalist [47], SUN [51], Places [58], and TEXTURE [4]. For the few-shot training, we follow the few-shot evaluation protocol adopted in CLIP [37] and CoOp [60], using 1, 2, 4, 8, and 16 shots for training, respectively, and deploying models in the full test sets. The average results over three runs are reported for comparison.

**Setup.** Following existing studies [46, 30, 33], we use ViT-B/16 [6] as a backbone. Specifically, we use the publicly available CLIP-ViT-B/16 models (https://github.com/openai/CLIP). The resolution of CLIP's feature map is $14 \times 14$ for CLIP-ViT-B/16. For $K$, we use 200 in all experiments. For $\lambda$, we use 0.25 in all experiments. The sensitivities of $K$ and $\lambda$ are described in the Analysis section 4 for $K$ and in the supplementary material for $\lambda$. Other hyperparameters (*e.g.*, training epoch=50, learning rate=0.002, batch size=32, and token lengths $N$=16) are the same as those of CoOp [60]. We use a single Nvidia A100 GPU for all experiments. Since LoCoOp takes approximately 1.4 times longer than CoOp, an additional experiment was conducted using 70 epochs for CoOp, but the results did not differ from those of 50 epochs. Therefore, we report all the results with 50 epochs.

**Comparison Methods.** To evaluate the effectiveness of our LoCoOp, we compare it with zero-shot detection methods, fully-supervised detection methods, and the baseline prompt learning method. For zero-shot OOD detection methods, we use MCM [30] and GL-MCM [33], which are state-of-the-art zero-shot detection methods. For fully-supervised detection methods, we follow the comparison methods [28, 50, 45, 46] in NPOS [46]. All the other baseline methods are fine-tuned using the same pre-trained model weights (*i.e.*, CLIP-B/16). For NPOS, the MCM score function is used in the original paper. For a fair comparison, we reproduce NPOS and also report the results with the GL-MCM score. For prompt learning methods, we use CoOp [60], which is the representative work and the baseline for our LoCoOp. As for the comparison with CoCoOp [61], which is another representative work, we show the results in Analysis Section 4.

Table 1: **Comparison results on ImageNet OOD benchmarks.** We use ImageNet-1K as ID. We use CLIP-B/16 as a backbone. Bold values represent the highest performance. † is cited from [46]. * is our reproduction. We find that LoCoOp with GL-MCM (LoCoOp$_{GL}$) is the most effective method.

| Method | iNaturalist FPR95↓ | iNaturalist AUROC↑ | SUN FPR95↓ | SUN AUROC↑ | Places FPR95↓ | Places AUROC↑ | Texture FPR95↓ | Texture AUROC↑ | Average FPR95↓ | Average AUROC↑ |
|---|---|---|---|---|---|---|---|---|---|---|
| *Zero-shot* | | | | | | | | | | |
| MCM [30]* | 30.94 | 94.61 | 37.67 | 92.56 | 44.76 | 89.76 | 57.91 | 86.10 | 42.82 | 90.76 |
| GL-MCM [33]* | 15.18 | 96.71 | 30.42 | 93.09 | 38.85 | 89.90 | 57.93 | 83.63 | 35.47 | 90.83 |
| *Fine-tuned* | | | | | | | | | | |
| ODIN [28]† | 30.22 | 94.65 | 54.04 | 87.17 | 55.06 | 85.54 | 51.67 | 87.85 | 47.75 | 88.80 |
| ViM [50]† | 32.19 | 93.16 | 54.01 | 87.19 | 60.67 | 83.75 | 53.94 | 87.18 | 50.20 | 87.82 |
| KNN [45]† | 29.17 | 94.52 | 35.62 | 92.67 | 39.61 | 91.02 | 64.35 | 85.67 | 42.19 | 90.97 |
| NPOS$_{MCM}$ [46]† | 16.58 | 96.19 | 43.77 | 90.44 | 45.27 | 89.44 | 46.12 | 88.80 | 37.93 | 91.22 |
| NPOS$_{MCM}$ [46]* | 19.59 | 95.68 | 48.26 | 89.70 | 49.82 | 88.77 | 51.12 | 87.58 | 42.20 | 90.43 |
| NPOS$_{GL}$* | 18.70 | 95.36 | 38.99 | 90.33 | 41.86 | 89.36 | 47.89 | 86.44 | 36.86 | 90.37 |
| *Prompt learning* | | | *one-shot (one label per class)* | | | | | | | |
| CoOp$_{MCM}$ | 43.38 | 91.26 | 38.53 | 91.95 | 46.68 | 89.09 | 50.64 | 87.83 | 44.81 | 90.03 |
| CoOp$_{GL}$ | 21.30 | 95.27 | 31.66 | 92.16 | 40.44 | 89.31 | 52.93 | 84.25 | 36.58 | 90.25 |
| LoCoOp$_{MCM}$ (ours) | 38.49 | 92.49 | 33.27 | 93.67 | 39.23 | 91.07 | 49.25 | 89.13 | 40.17 | 91.53 |
| LoCoOp$_{GL}$ (ours) | 24.61 | 94.89 | 25.62 | 94.59 | 34.00 | **92.12** | 49.86 | 87.49 | 33.52 | 92.14 |
| | | | *16-shot (16 labels per class)* | | | | | | | |
| CoOp$_{MCM}$ | 28.00 | 94.43 | 36.95 | 92.29 | 43.03 | 89.74 | **39.33** | 91.24 | 36.83 | 91.93 |
| CoOp$_{GL}$ | **14.60** | 96.62 | 28.48 | 92.65 | 36.49 | 89.98 | 43.13 | 88.03 | 30.67 | 91.82 |
| LoCoOp$_{MCM}$ (ours) | 23.06 | 95.45 | 32.70 | 93.35 | 39.92 | 90.64 | 40.23 | **91.32** | 33.98 | 92.69 |
| LoCoOp$_{GL}$ (ours) | 16.05 | **96.86** | **23.44** | **95.07** | **32.87** | 91.98 | 42.28 | 90.19 | **28.66** | **93.52** |

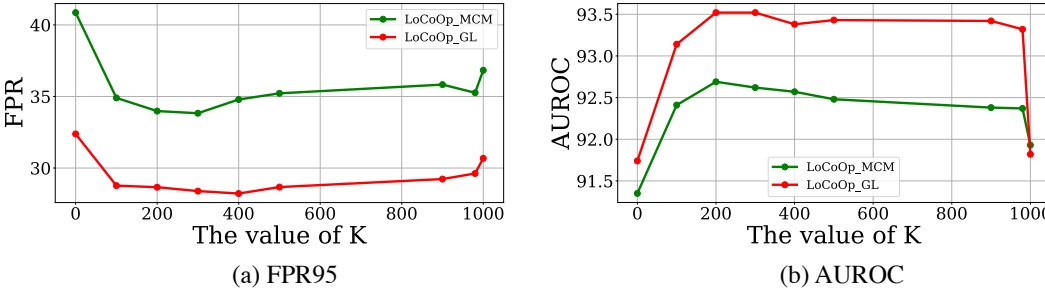

| (a) FPR95 | (b) AUROC |
|---|---|

Figure 3: Ablation studies on $K$. We report average FPR95 and AUROC scores on four OOD datasets in Table 1. The performance is not very sensitive to $K$ values except for extreme values such as 0 and 1,000.

**Evaluation Metrics.** For evaluation, we use the following metrics: (1) the false positive rate of OOD images when the true positive rate of in-distribution images is at 95% (FPR95), (2) the area under the receiver operating characteristic curve (AUROC).

## 3.2 Main results

The OOD detection performances are summarized in Table 1. We show that LoCoOp with GL-MCM (LoCoOp$_{GL}$) can achieve superior OOD detection performance, outperforming the competitive methods by a large margin. In particular, with a 16-shot setting, LoCoOp with GL-MCM reached an AUROC score of 93.52%, while the other methods failed to reach an AUROC score of 92%.

In addition, we can see, even in a one-shot setting, LoCoOp with GL-MCM achieved average FPR95 and AUROC scores of 33.52 and 92.14, respectively, which outperforms the zero-shot and fine-tuned methods. Considering that LoCoOp utilizes only one training sample per class while fine-tuned methods use 1,000 training images per class, these results are remarkable.

Table 2: Comparison results with CoCoOp [61]. We report average FPR and AUROC scores on four OOD datasets in Table 1.

| | | Average | |
|---|---|---|---|
| Method | Infer time↓ | FPR95↓ | AUROC↑ |
| CoCoOp$_{MCM}$ [61] | 149 ms | 35.53 | 91.99 |
| LoCoOp$_{MCM}$ | **2.59 ms** | 33.98 | 92.69 |
| LoCoOp$_{GL}$ | 5.97 ms | **28.66** | **93.52** |

Table 3: Comparison in ID accuracy on ImageNet-1K validation data.

| Method | Top-1 Accuracy |
|---|---|
| CoOp | **72.1** |
| LoCoOp | 71.7 |

In Fig. 2, we show the results with different numbers of ID labeled samples for CoOp and LoCoOp, where the green line denotes our LoCoOp with MCM, the red one denotes our LoCoOp with GL-MCM, the blue one is CoOp with MCM, and the orange line denotes CoOp with GL-MCM. The detailed accuracy can be found in the supplementary materials. We observed that GL-MCM boosts the performance of LoCoOp much more than CoOp in all settings.

### 3.3 Ablation studies on hyperparameter K

We conduct ablation studies on $K$ in Eq. (4). $K$ is an important parameter to pick up OOD regions from the local features. We experimented with a 16-shot setup using $K$=0 (all the local regions are used for OOD), 100, 200, 300, 400, 500, 900, 980, and 1,000 (same as CoOp). In Fig. 3, we summarize the results of ablation studies. First, let us look at the leftmost point of the graph, $K$=0, *i.e.*, the result when all the local regions are treated as OOD. We can see that $K$=0 degrades the detection performance in all settings. This is because it treats all object-relevant regions as OOD. Then, going around $K$=100, we can see that the performance improves dramatically because the number of such false negative OOD features is decreasing. LoCoOp maintains high performance after $K$=200. In particular, even at $K$=980, a large performance gap exists with $K$=1,000 (CoOp). At $K$=980, the number of OOD regions per image is about one, indicating that LoCoOp is effective even when only a few local OOD features are used for training. In general, the value of $K$ is not sensitive to the performance and is not difficult to select as long as it is not set to extreme values such as 0 or 1,000.

## 4 Analysis

**Comparison with CoCoOp.** We compare our LoCoOp with CoCoOp [61]. CoCoOp is also representative work for prompt learning, which generates an input-conditional text token for each image. For comparison, we add per-image inference time (in milliseconds, averaged across test images) for evaluation metrics [45]. This is because OOD detection methods are applied before test-time input comes to the close-set classifier, so the short inference time is important. We use a single Nvidia A100 GPU for the measurements. We show the 16-shot results of CoCoOp and LoCoOp in Table 2. This result demonstrates that our LoCoOp outperforms CoCoOp in performance and inference speed. Especially, CoCoOp takes a much longer inference time than our LoCoOp. That is because it requires an independent forward pass of instance-specific prompts for each image through the text encoder. When inferring ImageNet OOD benchmarks (total number of images is 85,640), CoCoOp takes 3.55 hours, while LoCoOp with MCM takes only 3.7 minutes.

**ID accuracy of LoCoOp.** We examine the ID accuracy of LoCoOp. OOD detection is a preprocessing before input data comes into the ID classifier, and its purpose is to perform binary classification of ID or OOD [15]. However, it is also important to investigate the relationship between close-set ID accuracy and OOD detection performance [49]. In Table 3, we show the 16-shot ID accuracy of CoOp and LoCoOp on the ImageNet-1K validation set. This result shows that LoCoOp is slightly inferior to CoOp in ID accuracy. We hypothesize that excluding OOD nuisances that are correlated with ID objects will degrade the ID accuracy. For example, in some images of dogs, the presence of green grass in the background may help identify the image as a dog. Therefore, learning to remove the background information could make it difficult to rely on such background information to determine that the image is a dog. However, this study demonstrates that excluding such backgrounds improves

Table 4: Comparison results with CLIP-ResNet-50. We find that LoCoOp with GL-MCM (LoCoOp$_{GL}$) is effective with ResNet-50.

| Method | iNaturalist FPR95↓ | AUROC↑ | SUN FPR95↓ | AUROC↑ | Places FPR95↓ | AUROC↑ | Texture FPR95↓ | AUROC↑ | Average FPR95↓ | AUROC↑ |
|---|---|---|---|---|---|---|---|---|---|---|
| MCM [30] | 31.98 | 93.86 | 46.09 | 90.75 | 60.56 | 85.67 | 60.00 | 85.72 | 49.66 | 89.00 |
| GL-MCM [33] | 37.20 | 90.98 | 38.74 | 91.21 | 49.49 | 86.40 | 43.46 | 88.14 | 42.22 | 89.18 |
| CoOp$_{GL}$ | 33.68 | 92.59 | 40.0 | 90.77 | 48.14 | 86.8 | **31.27** | 91.69 | 38.27 | 90.46 |
| LoCoOp$_{GL}$ (ours) | **22.81** | **95.38** | **36.44** | **92.22** | **44.10** | **88.3** | 34.47 | **91.89** | **34.45** | **91.95** |

Table 5: Comparison with other methods for ID-irrelevant regions extraction.

| Method | iNaturalist FPR95↓ | AUROC↑ | SUN FPR95↓ | AUROC↑ | Places FPR95↓ | AUROC↑ | Texture FPR95↓ | AUROC↑ | Average FPR95↓ | AUROC↑ |
|---|---|---|---|---|---|---|---|---|---|---|
| Ent. | 25.05 | 94.28 | 31.92 | 92.97 | 39.85 | 89.94 | 48.71 | 87.52 | 36.38 | 91.18 |
| Prob. | 16.19 | 96.55 | 23.66 | 94.76 | **32.62** | 91.83 | **42.12** | 89.96 | **28.65** | 93.27 |
| Rank (ours) | **16.05** | **96.86** | **23.44** | **95.07** | 32.87 | **91.98** | 42.28 | **90.19** | 28.66 | **93.52** |

the OOD detection performance. This study exhibits that a strong ID classifier is not always a robust OOD detector for prompt learning.

**Effectiveness with CNN architectures.** We examine the effectiveness of LoCoOp with CNN architectures. In the main experiments, we use a Vision Transformer architecture as a backbone following existing work [46, 30]. However, for the fields of OOD detection, CNN-based models also have been widely explored [17, 43]. Following this practice, we use ResNet-50 as a CNN backbone and examine the effectiveness of LoCoOp. Table 4 shows the 16-shot results with ResNet-50. We find that similar to the results with ViT, our LoCoOp is effective for ResNet-50.

**Comparison with other methods for ID-irrelevant region extraction.** We explore the effectiveness of several methods for ID-irrelevant region extraction. Although we use a Top-$K$ rank-based thresholding approach, two other approaches, entropy-based thresholding and probability-based thresholding, approaches are also candidates for the methods. Entropy-based thresholding methods use the threshold of the entropy of $p_i$, whereas probability-based thresholding methods use the threshold of the value of $p_i(y = y^{in}|x)$. For entropy-based thresholding methods, we extract regions where the entropy of $p_i$ is less than $\frac{\log M}{2}$ because $\frac{\log M}{2}$ is the half value of the entropy of $M$-dimmensional probabilities, which is following existing studies [40, 55]. For probability-based thresholding methods, we extract regions where $p_i(y = y^{in}|x^{in})$ is less than $1/M$ simply because the ID dataset has $M$ classes. We summarize the 16-shot OOD detection results in Table 5. For the entropy-based method, the performance is poor since it is challenging to determine the appropriate threshold. On the other hand, for the probability-based method, the performance is comparable to our rank-based method since we can determine the threshold intuitively. However, we consider that, from the user's perspective, the rank is easier to interpret and handle than the probability value. Therefore, we use rank-based thresholding in this paper.

**Visualization of extracted OOD regions.** In Fig.4, we show visualization results of extracted OOD regions. The performance of OOD extraction is key to our method. From this visualization result, we find that our Rank-based approach can accurately identify OOD regions.

## 5 Theoretical background

The effectiveness of separating foregrounds (ID objects) and backgrounds is theoretically guaranteed by existing research [31, 39]. Ming *et al.*[31] states that when the classifier only uses foreground features, the optimal decision boundary can be obtained. With conventional classifiers, it was difficult for the classifiers to use only foreground features. Contrary to this, CLIP has rich local visual-text alignments and can identify the foregrounds and backgrounds. Based on the theoretical analysis in [31], LoCoOp achieves high OOD detection performance.

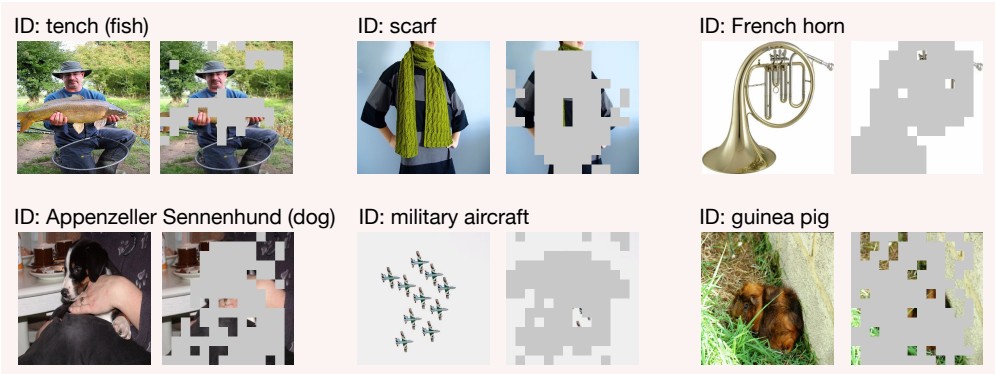

Figure 4: **Visualization of extracted OOD regions.** We find that our approach can correctly extract ID-irrelevant regions.

# 6 Limitations

**Application to other light-weight tuning methods.** In this study, we focus on text prompt learning methods [60, 61] because they are the most representative methods for tuning CLIP. These days, other light-weight tuning methods such as Tip-Adapter, which learns a lightweight residual feature adapter [57] and visual prompt methods [21, 2], which learn a visual prompt added to the input image [2] or inside the layer [21], have been proposed for image classifications. It is our future work to apply our LoCoOp to these methods [57, 21, 2].

**Application to the model without the rich local visual-text alignment.** This approach requires models with strong local visual-text alignment capabilities, such as the image encoder in CLIP. Consequently, applying this approach to models lacking such capabilities can prove challenging. Nonetheless, given the numerous methods currently emerging based on CLIP, our research is influential in this research field.

**Extending to other visual recognition tasks.** In this study, we focus on only classification tasks. However, prompt learning is widely used for other tasks, such as object detection [10] and segmentation [10]. We will solve other tasks as important future work. However, existing studies about OOD detection do not deal with both classification and other tasks simultaneously [8, 26, 7], and most studies in OOD detection address classification tasks [53]. Therefore, we believe that our findings in the classification task are influential.

# 7 Related Work

**Out-of-distribution detection.** Various approaches have been developed to address OOD detection in computer vision [14, 23, 34–36, 15, 17, 19, 28, 45, 29, 26, 50, 53, 56, 32]. One of the common OOD detection approaches is to devise score functions such as probability-based method [15, 17, 19, 28, 44, 45], logit-based method [29, 17, 44], and feature-based method [26, 50]. Another approach is to leverage OOD data for training-time regularization [16, 54, 9, 52]. These OOD detection methods assume the use of backbones trained with single-modal data. In recent years, much attention has been paid to how multimodal pre-trained models, such as CLIP, can be used for out-of-distribution detection. Existing CLIP-based OOD detection methods have explored the zero-shot methods [12, 11, 30, 33] and fine-tuned method [46]. However, these methods have their own limitations regarding performance and training efficiency. To bridge the gap between fully supervised and zero-shot methods, developing a few-shot OOD detection method that utilizes a few ID training images for OOD detection is essential. Before CLIP emerged, Jeong *et al.* [20] addressed the few-shot OOD detection, using only a few ID training data for OOD detection, but their experiments have only been done on small data sets (*e.g.*, Omniglot [25], CIFAR-FS [3]), showing the limitations of the performances of few-shot learning methods without CLIP. By leveraging the powerful representations of CLIP, few-shot OOD detection could provide a more efficient and effective solution.

**Prompting for foundation models.** Prompting is a heuristic way to directly apply foundation models to downstream tasks in a zero-shot manner. Prompt works as a part of the text input that instructs the model to perform accordingly on a specific task. However, such zero-shot generalization relies heavily on a well-designed prompt. Prompt tuning [27] proposes to learn the prompt from downstream data in the continual input embedding space, which is a parameter-efficient way of fine-tuning foundation models. Although prompt tuning was initially developed for language models, it has been applied to other domains, including vision-language models [60, 61, 42, 38, 13]. CoOp [60], which applies prompt tuning to CLIP, is a representative work that effectively improves CLIP's performance on corresponding downstream tasks. However, existing prompt learning methods focus on improving only close-set ID classification accuracy, and their performance for OOD detection remains unclear. In this work, we propose a novel method for prompt learning for OOD detection.

**Local features of CLIP.** Recent work [59] revealed that CLIP has local visual features aligned with textual concepts. For classification tasks, global features are used, which are created by pooling the feature map with a multi-headed self-attention layer [48]. However, these features are known to lose spatial information [41], so they are not suitable for tasks that require spatial information, such as semantic segmentation. To adopt CLIP to segmentation, Zhou *et al.* [59] proposed the CLIP's local visual features aligned with textual concepts, which can be obtained by projecting the value features of the last attention layer into the textual space. Existing studies used CLIP's local features for semantic segmentation [59, 62, 22], multi-label classification [42] and ID detection [33], which need to detect local objects in an image. Unlike previous studies, we use local features to pick up ID-irrelevant parts (*e.g.*, backgrounds), which can be regarded as OOD features for OOD detection.

## 8   Conclusion

We present a novel vision-language prompt learning approach for few-shot OOD detection. To solve this problem, we present LoCoOp, which performs OOD regularization during training using the portions of CLIP local features as OOD images. CLIP's local features have a lot of ID-irrelevant information, and by learning to push them away from the ID class text embeddings, we can remove the nuisances in the ID class text embeddings, leading to a better separation of ID and OOD. Experiments on the large-scale ImageNet OOD detection benchmark demonstrate the advantages of our LoCoOp over zero-shot, fully supervised detection methods and existing prompt learning methods.

**Broader Impact.** This work proposed a prompt learning approach for few-shot OOD detection, improving the performance and efficiency of existing detection methods. Our work reduces the data-gathering effort for data-expensive applications, so it makes the technology more accessible to individuals and institutions that lack abundant resources. However, it is important to exercise caution in sensitive applications as our approach may make these systems vulnerable to misuse by individuals or organizations seeking to engage in fraudulent or criminal activities. Therefore, human oversight is necessary to ensure that our approach is used ethically and not relied upon solely for making critical decisions.

## Acknowledgement

This work was partially supported by JST JPMJCR22U4, CSTI-SIP, and JSPS KAKENHI 21H03460.

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
