# OpenReview forum: "LoCoOp: Few-Shot Out-of-Distribution Detection via Prompt Learning"
_NeurIPS.cc/2023/Conference — NeurIPS 2023 poster_

### Official Review · Reviewer_epH7 · 2023-07-01

**Soundness:** 3 good
**Presentation:** 4 excellent
**Contribution:** 3 good
**Rating:** 5
**Confidence:** 4

**Summary:**

The paper proposed locally regularized Context Optimization for OOD detection inspired by CoOP. Their primary claim is that the CLIP feature contains many ID-irrelevant nuances, such as backgrounds. Hence pushing the ID and these embedding from each other will lead to a better separation between ID and OOD samples. It is a few shot OOD detection methods that utilize learnable prompts. The author conducted experiments on zero/one/few shot methods, which shows their proposed method outperforms existing approaches.

**Strengths:**

The strengths of the paper are :
1. The paper is well-written and easy to follow.
2. It is a simple approach combined with CoOP, MCM, and GL-MCM scores for test time prediction, giving 3. SOTA performance in zero-shot settings.
4. Pushing apart the ID-irrelevant features from CLIP, similar to MaskCLIP, would lead to a better OOD classification is a novel idea and has much potential.
5. The results are also good; it outperforms most existing models using both ResNet and ViT.


**Weaknesses:**

There are a small few weaknesses of the paper which the author needs to address  :
1. The prompt learning used here is the same as CoOP, then why is it novel?
2. Table 1. Zero-shot uses CLIP or CoOP, it is not clear.
3. The performance using CoOP with MCM/GL-MCM is already quite good, and LoCoOP improves an additional 1-2% gain in AUROC only in the one-shot or few-shot setting. Hence I doubt its efficacy in large-scale OOD detection or in near OOD detection.

**Questions:**

Are there any results on near OOD or LargeScale datasets?

**Limitations:**

No negative social impact and limitations are addressed adequately

---

> ### Author Rebuttal · Authors · 2023-08-08
>
> We thank the reviewer for the valuable comments and respond to them appropriately as follows. We will add suggested experiments and explanations in the updated manuscript.
>
> > Q1. The novelty of LoCoOp
>
> A1. While CoOp learns to bring the global image features and GT text feature closer together, LoCoOp treats the portions of CLIP's local features as OOD and performs OOD regularization in addition to the CoOp's loss. Although it seems like just adding an OOD regularization term to CoOp, it is very effective for OOD detection. Therefore, we consider that LoCoOp has a solid novelty for few-shot OOD detection.
>
> > Q2. Zero-shot method
>
> A2.  We use CLIP for zero-shot results.
>
> > Q3. Performance improvements with LoCoOp
>
> A3. That is true that LoCoOp brings 1-2% gain in AUROC over CoOp, but we argue that this improvement is not small on ImageNet-1K OOD benchmarks. Existing work (NPOS [43]) reported that NPOS brings only a 0.03 % gain over VOS [9] on ImageNet-1K benchmarks. Therefore, we consider a 1~2% gain to be very significant in this tight setting where we have only a few labels data with the large ImageNet benchmarks.

---

> > ### Comment · Area_Chair_wwZH · 2023-08-15
> >
> > Dear reviewer epH7, we are in the middle of the discussion period. Please read the rebuttal and use the time to discuss with the authors.

---

> > > ### Comment · Reviewer_epH7 · 2023-08-22
> > >
> > > Thank you for your response. While the authors try to address the concern, the underlying prompt are precisely the same. And in OOD detection, the most difficult thing is to detect near-OOD, which the authors didn't provide any information in the rebuttals. So, I will keep my rating unchanged.

---

> ### Author Response · Authors · 2023-08-19
> **Looking Forward to Hearing from Reviewer epH7**
>
> Dear reviewer, we would like to thank you again for your careful review and constructive suggestions. We have provided additional explanations in response to your concerns.
>
> As the deadline for the author-reviewer discussion period is approaching, we would like to discuss with you whether your concerns have been addressed. And if you have any additional comments, we are happy to answer them further.

---

### Official Review · Reviewer_RaSR · 2023-07-02

**Soundness:** 4 excellent
**Presentation:** 4 excellent
**Contribution:** 4 excellent
**Rating:** 7
**Confidence:** 4

**Summary:**

This paper focused on the problem of vision-language prompt learning for few-shot OOD detection, i.e., using CLIP model to detect OOD images from unseen classes using only a few labeled in-distribution (ID) images. Previous zero-shot methods may encounter a domain gap with ID downstream data, while fully fine-tuned methods require enormous training costs. This work focused on the few-shot learning setting and extends CoOp, a few-shot vision-language prompt learning, to OOD detection. The main difference is to extract ID-irrelevant regions based on region-class similarity for OOD regularization. Experiments are conducted on iNaturalist, SUN, Places, and TEXTURE as OOD datasets and take ImageNet-1K as ID data. Experimental results verified the effectiveness of the proposed LoCoOp, even performing better than zero-shot and fully fine-tuned methods with 1-shot training data.

**Strengths:**

+ The problem of vision-language prompt learning for few-shot OOD detection is novel and interesting. Previous works focused on zero-shot or fully fine-tuning settings, while few-shot OOD detection is not well explored. Also, it makes sense to adapt few-shot prompt tuning method CoOp for the few-shot OOD detection.

+ The proposed method is simple but effective and intuitive. It directly extract pseudo OOD samples from few-shot ID samples. The motivation and solution are clear and inspiring.

+ The experimental results verified the effectiveness of the proposed method. Especially, it can perform better than zero-shot or fine-tuned methods even with one-shot training data. Compared to complex method CoCoOp, it can also achieve faster inference speed.

+ The writing quality is satisfying. It is easy to follow the idea and implementation.

**Weaknesses:**

- Although Table 3 compares the ID accuracies of CoOp and LoCoOp, it is interesting to know how other zero-shot and fine-tuned methods perform on ID accuracy, as CoOp may not achieve good ID performance due to the limitation of sample numbers.

- As the motivation is to extract OOD regions from ID images, it would be interesting to visualize the OOD regions to illustrate how well CLIP can output OOD regions.


------------------

After rebuttal:

Thank the authors for providing further comparisons and discussions. The comparisons and discussions are comprehensive. I will keep my positive rating.

**Questions:**

Please see weaknesses for my quesions.

**Limitations:**

The authors have adequately addressed the limitations

---

> ### Author Rebuttal · Authors · 2023-08-08
>
> We thank the reviewer for the valuable comments and respond to them as follows. We will add suggested experiments and explanations in the updated manuscript.
>
> > Q1. The ID accuracy with zero-shot and fully-supervised methods
>
> A1. The results of the ID accuracy and OOD performance on the ImageNet validation set for the zero-shot, fully supervised, and prompt-based methods are shown below.
>
> |  | ID acc. (%) | AUROC (%) |
> | --- | --- | --- |
> | Zero-shot (CLIP [35])  | 67.01 | 90.83 |
> | Fine-tune (NPOS [43]) | 79.42 | 90.37 |
> | CoOp [56] | 72.10 | 91.82 |
> | LoCoOp (ours) | 71.70 | 93.52 |
>
> Even though MCM, LoCoOp, and CoOp are inferior to fully supervised methods in ID accuracy, they performed better in OOD detection.
>
> > Q2. Visualization of OOD regions
>
> A2. Thanks for the suggestion.  We attached visualization results for some samples to demonstrate the effectiveness of our method in the global response. This shows that our method can correctly extract OOD regions.

---

> > ### Comment · Area_Chair_wwZH · 2023-08-15
> >
> > Dear reviewer RaSR, we are in the middle of the discussion period. Please read the rebuttal and use the time to discuss with the authors.

---

> > ### Comment · Reviewer_RaSR · 2023-08-15
> > **ID and OOD accuracies in the same table**
> >
> > Thank the authors for the feedback. For the ID accuracy, could the authors show a new table combining both ID and OOD accuracies for a more clear comparison? It would be more straightforward to see how LoCoOp decreases ID accuracy and increases OOD accuracy, and whether it achieves a great trade-off or simply increases one by decreasing the other to the same extend.

---

> > > ### Author Response · Authors · 2023-08-16
> > > **Relationship between ID accuracy and OOD detection performance**
> > >
> > > Thanks for the suggestion. We show a new table combining both OOD detection performance (Table 1 in the main paper) and ID accuracies in the following. The column on ID accuracy is on the far right.
> > >
> > > |  | iNaturalist |  | SUN |  | Places |  | Texture |  | Average |  | **ID acc.**  |
> > > | --- | --- | --- | --- | --- | --- | --- | --- | --- | --- | --- | --- |
> > > |  | FPR | AUROC | FPR | AUROC | FPR | AUROC | FPR | AUROC | FPR | AUROC |- |
> > > | Zero-shot |  |  |  |  |  |  |  |  |  |  |  |
> > > | MCM | 30.94 | 94.61 | 37.67 | 92.56 | 44.76 | 89.76 | 57.91  | 86.10 | 42.82 | 90.76 | 67.01 |
> > > | GL-MCM | 15.18 |  96.71  | 30.42  | 93.09  | 38.85  | 89.90  | 57.93  | 83.63 | 35.47 | 90.83 | 67.01 |
> > > | Fine-tune |  |  |  |  |  |  |  |  |  |  |  |
> > > | ODIN | 30.22  | 94.65  | 54.04  | 87.17  | 55.06  | 85.54 | 51.67  | 87.85  | 47.75  | 88.80 | 79.64 |
> > > | ViM | 32.19  | 93.16  | 54.01  | 87.19  | 60.67  | 83.75 | 53.94 | 87.18 | 50.20 | 87.82 | 79.64 |
> > > | KNN | 29.17 | 94.52  | 35.62 | 92.67  | 39.61 | 91.02  | 64.35 | 85.67 | 42.19 | 90.97 | 79.64 |
> > > | NPOS | 16.58 | 96.19 | 43.77 | 90.44 | 45.27 | 89.44 | 46.12 | 88.80 | 37.93 | 91.22 | 79.42 |
> > > | Prompt learning |  |  |  |  |  |  |  |  |  |  |  |
> > > | CoOp w. MCM(1-shot) | 43.38 | 91.26 | 38.53 | 91.95 | 46.68 | 89.09  | 50.64 | 87.83 | 44.81  | 90.03 | 66.23 |
> > > | CoOp w. GL-MCM(1-shot) | 21.30 | 95.27 | 31.66 | 92.16 | 40.44 | 89.31 | 52.93 | 84.25 | 36.58 | 90.25 | 66.23 |
> > > | LoCoOp w. MCM (1-shot) | 38.49 | 92.49 | 33.27 | 93.67 | 39.23 | 91.07 | 49.25  | 89.13  | 40.17 | 91.53 | 66.88 |
> > > | LoCoOp w. GL-MCM (1-shot) | 24.61 | 94.89 | 25.62 | 94.59 | 34.00 | 92.12 | 49.86 | 87.49 | 33.52 | 92.14 | 66.88 |
> > > | CoOp w. MCM(16-shot) | 28.00 | 94.43 | 36.95 | 92.29 | 43.03 | 89.74 | 39.33 | 91.24 | 36.83 | 91.93 | 72.10 |
> > > | CoOp w. GL-MCM(16-shot) | 14.60  |  96.62 | 28.48 | 92.65 | 36.49 | 89.98 | 43.13 | 88.03 | 30.67 | 91.82 | 72.10 |
> > > | LoCoOp w. MCM (16-shot) | 23.06 | 95.45 | 32.70 |  93.35 | 39.92 | 90.64  | 40.23 | 91.32 | 33.98 | 92.69 | 71.70 |
> > > | LoCoOp w. GL-MCM (16-shot) | 16.05 | 96.86 | 23.44 | 95.07 | 32.87 | 91.98 | 42.28 | 90.19 | 28.66 | 93.52 | 71.70 |
> > >
> > > ---
> > >
> > > We discuss the relationships between ID accuracy and OOD detection performance in the following three points.
> > >
> > > - Why zero-shot and prompt learning methods outperform fully-supervised methods in OOD detection performance while their ID accuracies are considerably lower.
> > >
> > > The key point in OOD detection is to avoid incorrectly assigning a high confidence score to OOD samples. In this respect, zero-shot and prompt learning methods calculate confidence scores based on the similarity between the text and the image, so models are less likely to produce unnaturally high confidence scores for OOD samples. On the other hand, most fully-supervised methods do not use the language, and use the probability distribution through the last fc layer to calculate confidence scores. Therefore, even if the ID accuracy is high, there is a higher possibility that the model will produce an incorrect high confidence score for an OOD sample due to some reasons (e.g., noisy activation signal [40]).
> > >
> > > - Why LoCoOp has higher ID accuracy than CoOp in a 1-shot setting
> > >
> > > This is because CoOp does not have enough training samples in a 1-shot setting. As shown in Fig. 2, CoOp and LoCoOp require about 16-shot image-label pairs to reach the upper score.
> > >
> > > On the other hand, even in a 1-shot setting, LoCoOp can learn from many OOD features, so  LoCoOp outperforms CoOp in ID accuracy in a 1-shot setting.
> > >
> > > -  Why LoCoOp has lower ID accuracy than CoOp in a 16-shot setting
> > >
> > > This reason is described in Analysis section in the main paper.
> > > In a 16-shot setting (sufficient training data for prompting methods), excluding OOD nuisances that are correlated with ID objects will degrade the ID accuracy. For example, in some images of dogs, the presence of green grass in the background may help identify the image as a dog. Therefore, learning to remove the background information could make it difficult to rely on such background information to determine that the image is a dog. However, this study reveals that excluding such backgrounds improves OOD detection performance.
> > >
> > >
> > > As the reviewer says, it is intriguing to discuss the relationship between ID accuracy and OOD detection performance. On the other hand, we are concerned that incorporating ID accuracy into Table 1 would increase the amount of information to be discussed in a single table.
> > > Therefore, in the final version, I will create another section discussing the relationship between ID accuracy and OOD detection performance and include the above table with ID accuracy and detection performance.

---

> > > > ### Comment · Reviewer_RaSR · 2023-08-16
> > > > **The discussion between ID accuracy and OOD detection performance is good**
> > > >
> > > > Thank the authors for providing further comparisons and discussions. The comparisons and discussions are comprehensive. I agreed that a single table to include all things would be a burden, and the discussion could be added to the appendix. I currently have no more concerns.

---

### Official Review · Reviewer_wEUW · 2023-07-05

**Soundness:** 2 fair
**Presentation:** 3 good
**Contribution:** 2 fair
**Rating:** 5
**Confidence:** 4

**Summary:**

In this paper, the authors propose a CLIP-based few-shot OOD detector named Local regularized Context Optimization (LoCoOp). The method LoCoOp uses learnable prompts and local tokens' class scores to optimize the OOD detection performance. During the few-shot training phase, the authors extract the ID-irrelevant regions by top-k ranking w.r.t. the ground truth class and minimize their class entropy. In this way, the affinities between local tokens and ID classes are suppressed, thus enhancing the distinction between ID and OOD samples. The experimental results show the proposed method outperforms baseline methods.

**Strengths:**

1. The paper is well-organized and clearly presented. The proposed method is easy to follow.
2. LoCoOp is a novel method that uses OOD region regularization to suppress the ID similarity of the background components. The idea of ID-irrelevant local context suppression is interesting. The proposed method raises a valuable point that suppressing extraneous background helps distinguish ID from OOD data.

**Weaknesses:**

1. The method needs more analysis.
   - The authors may provide some visualizations for ID and OOD samples to show which regions are suppressed. This kind of case study helps demonstrate that the proposed method works as the authors envision.
   - The foreground object F and its background context B may tend to co-occurrence, so there may be a spurious correlation issue [1]. Hence, the GT label may also rank higher in the background context. For example, for the background, like the beach, categories associated with the beach (e.g., beach chairs, surfboards) may rank higher than categories that have no relations. Based on this, for some categories with spurious correlation, will this training strategy fail? Because the ground-truth label may occur in top-k candidates for almost all background regions.
   - The OOD regularization loss (Eqn. 5) needs to be further analyzed. The authors may investigate the training dynamics of the regularization loss. How does this loss affect prompt learning convergence? Will it cause instability and affect ID accuracy in a longer non-few-shot training schedule? How does it affect hard samples?
   - The proposed method may need some theoretical analysis and support.
2. More experiments and validation are necessary.
   - I suggest that the authors extensively verify the effectiveness of the proposed method on various OOD benchmarks. In particular, it is necessary to pay attention to whether the proposed OOD regularization loss will behave as expected when the network is trained with fewer categories (such as CIFAR10)?
   - The authors only train the network with up to 16 samples per class. I'm curious if one can achieve better results by using more samples to finetune the network? Where is the bottleneck of OOD detection based on prompt learning? Can we unfreeze more layers to obtain stronger OOD regularization capabilities?
   - In Table 1, why does one-shot prompt learning of CoOp+MCM result in lower performances compared to zero-shot MCM?

[1] Ming, Yifei, Hang Yin, and Yixuan Li. "On the impact of spurious correlation for out-of-distribution detection." *Proceedings of the AAAI Conference on Artificial Intelligence*. Vol. 36. No. 9. 2022.

**Questions:**

Some minor issues.
   - The concept *OOD image features* in Line 146 is prone to ambiguity. Readers may think that it is a feature taken from the OOD image.
   - There may be a typo in Figure 2 x-axis label *# of  labels*. Perhaps *# of samples* is the correct one?

---

> ### Author Rebuttal · Authors · 2023-08-08
>
> We thank the reviewer for the valuable comments and respond to them as follows. We will add suggested experiments and explanations in the updated manuscript.
>
> > Q1. Visualization results
>
> A1. Thanks for pointing this out. We attached the visualization results. From this result, we can see the OOD regions are extracted correctly.
>
> > Q2. Spurious correlation issue
>
> A2. Thanks for the interesting point of view.  We consider that the spurious correlation issue does not occur in the CLIP’s local features. Unlike global features, CLIP's local features are features before pooling, so the feature in each region has no mixed concepts. In other words, areas with beaches have the concept of beaches only, and areas with beach chairs have the concept of beach chairs only. Therefore, ground-truth labels are unlikely to be included in the top-k candidates in background regions. This may be more readily understood by examining the visualization results that I have attached. The unmasked areas in these visualization examples are areas where the ground-truth labels are not included in the top-200 prediction. Therefore, we consider that spurious correlation does not occur in the local features.
>
> Based on this observation, in our view, LoCoOp can directly solve the problem of the spurious correlation issue.  LoCoOp separates the background from the foreground, which results in the elimination of spurious correlations between things that are normally correlated, such as beach and beach chairs. We will document this discussion in detail in the final version.
>
> > Q3. Analysis of the OOD regularization loss
>
> A3. The training convergence and instability for LoCoOp are similar to those of CoOp, and the training config is the same as that of CoOp.
>
> > Q4. Theoretical analysis and support.
>
> A4. Thanks for the important suggestion. In terms of the effectiveness of disentangling background information, our work is supported by existing theoretical studies [1, 2, 3], which state, when the classifier only uses foreground features, the optimal decision boundary can be obtained. Therefore, we will cite these theoretical studies [1, 2, 3] to reinforce the theoretical background in the final version.
>
> [1] Ren et al., Likelihood Ratios for Out-of-Distribution Detection, NeurIPS 2019.
> [2] Nagarajan et al., Understanding the Failure Modes of Out-of-Distribution Generalization, ICLR 2021.
> [3] Ming et al., On the impact of spurious correlation for out-of-distribution detection, AAAI2022.
>
> > Q5. The effectiveness of LoCoOp on small-scale datasets
>
> A5. As for the small-scale dataset, there are no existing studies dealing with CIFAR-10 in CLIP, because CLIP is not compatible with toy datasets (e.g., too low resolution) like CIFAR-10.   Hence, we experiment with an ImageNet subset dataset.
>
> Previous work [30] reported that the result on ImageNet-10 [30] (a 10-class subset of ImageNet) and ImageNet-20 [30] (a 20-class subset of ImageNet) with zero-shot MCM reached the upper-bound score (e.g., average AUROC is 99.78 on ImageNet-10). Therefore, we use ImageNet-100  (a 100-class subset of ImageNet) as the ID dataset. As for the OOD datasets, we adopt the same ones as the ImageNet-1K OOD datasets.
>
> The result on ImageNet100 OOD datasets is as follows.
>
> |  | FPR (%) | AUROC (%) |
> | --- | --- | --- |
> | CoOp with MCM | 14.57 | 97.12 |
> | CoOp with GL-MCM | 13.82 | 96.93 |
> | LoCoOp with MCM (ours) | 12.68 | 97.49 |
> | LoCoOp with GL-MCM (ours) | **10.77** | **97.67** |
>
> From this result, LoCoOp outperforms CoOp on small-scale datasets.
>
> > Q6. The performance of LoCoOp with more than 16-shot samples
>
> A6. In this paper, we conducted experiments up to 16 shots following the setting of CoOp.  Figure 2 shows that the improvement in performance will become slower after 16-shot training. This is due to the fact that CoOp and LoCoOp only train text prompts that have only a few parameters, and a small number of samples is sufficient to reach the upper score.
>
> A key element to further improve performance is CLIP's vision encoder. LoCoOp freezes the vision encoder, so the image features may contain some OOD information. Removing OOD information in image features can be a possible solution to further OOD detection performance.
>
> > Q7. The lower performance of 1-shot CoOp with MCM than that of zero-shot MCM
>
> A7. According to the paper in CoOp, the classification accuracy of prompt learning methods with 1-shot training is inferior to the zero-shot performance because there are not enough samples to learn enough discriminative representations in prompts. Similarly, in the case of OOD detection, CoOp cannot learn enough features with only one sample. However, LoCoOp is capable of learning multiple OOD features from a single image. Therefore, even in a setting with only one image, it can learn a sufficient amount of features to achieve robust performance in OOD detection.
>
> > Minor issues: ambiguous description and  typo
>
> Thanks for the point out. We will fix them in the final version.

---

> > ### Comment · Area_Chair_wwZH · 2023-08-15
> >
> > Dear reviewer wEUW, we are in the middle of the discussion period. Please read the rebuttal and use the time to discuss with the authors.

---

> > ### Comment · Reviewer_wEUW · 2023-08-20
> >
> > Sorry for the late reply. The authors' responses address most of my concerns. I decided to raise my rating to 5. I still suggest that the author strengthen the theoretical modelling of the article.

---

> ### Author Response · Authors · 2023-08-19
> **Looking Forward to Hearing from Reviewer wEUW**
>
> Dear reviewer, we would like to thank you again for your careful review and constructive remarks. We have provided additional explanations and experiments in response to your concerns.
>
> As the deadline for the author-reviewer discussion period is approaching, we would like to discuss with you whether your concerns have been addressed. And if you have any additional comments, we are happy to answer them further.

---

### Official Review · Reviewer_uDwi · 2023-07-06

**Soundness:** 3 good
**Presentation:** 3 good
**Contribution:** 3 good
**Rating:** 5
**Confidence:** 4

**Summary:**

The task of this article is to use the CLIP model for image classification, and the target scene is few-shot data and out-of-distribution (OoD). The author confirms the final OoD region by querying the ID-independent region in the image, and further optimizes the model through this region. Remarkably, the authors validated their method on the large-scale dataset Image-Net 1K. As far as I know the task is a new one, and its application makes sense.

**Strengths:**

- This paper proposed a new and meaningful task.
- The performance of the method is fine.
- The author completed method verification on a very challenging dataset (ImageNet-1K full dataset).
- It's an interesting idea to remove regions not related to IDs.
- The author provides the source code to ensure the reproducibility of the article.

**Weaknesses:**

The idea of this article is very interesting and theoretically feasible. However, I have the following concerns:

- My biggest concern is the accuracy of locating regions not related to IDs. Because the author did not provide an experimental indicator of the accuracy of the ID-independent region positioning, it is only shown from the results of training and detecting OoD. My concerns come from another article I've read that points to the limitations of CLIP's current interpretation of location regions [1]. It is a pity not to see related visualization results in the article and supplementary material. If the author can provide some indicators to prove that his method can well locate regions that have nothing to do with ID, I think it will be more convincing.
- In addition to OoD detection, the model should have Zero-Shot capability, should the author also compare the zero-shot performance with CoOp or CoCoOp methods?

[1] Li, Yi, et al. "CLIP surgery for better explainability with enhancement in open-vocabulary tasks." *arXiv preprint arXiv:2304.05653* (2023).

**Questions:**

My main questions are in the weaknesses section, and I would like to improve my score if the author can address my concerns convincingly.

In addition, I still want to know how long (how many hours or how many days) does this method take to train the ImageNet dataset on an A100GPU?

**Limitations:**

Yes, the authors have listed the limitation in the article.

---

> ### Author Rebuttal · Authors · 2023-08-08
>
> We thank the reviewer for the valuable comments and respond to them as follows. We will add suggested experiments and explanations in the updated manuscript.
>
> > Q1. The accuracy of locating regions not related to IDs.
>
> A1. Thanks for the question.  That is correct, and the accuracy of the segmentation results is a major factor. However, unlike the usual segmentation task, in this setting, it is not necessary to correctly guess the segmentation label. Therefore, to compensate for errors in the segmentation results, we introduce ranks into equation (4) to allow for segmentation errors up to top-K predictions.
>
> As for the evaluation metrics, the ImageNet dataset does not have segmentation masks, so quantitative evaluation is difficult. Therefore, we have attached visualization results for some samples in the global response to demonstrate the effectiveness of our method. This shows that our method can correctly extract OOD regions.
>
> > Q2. Zero-shot generalization ID accuracy
>
> A2. We report the zero-shot generalization ID accuracy (not OOD detection performance) with CoOp and LoCoOp on Flower102, Food101, and Oxford-Pets datasets. Note that LoCoOp aims to improve the performance of OOD detection on specific ID datasets, so the zero-shot generalization ID accuracy is not objective.
> | Method      | Flower | Food  | Pets  |
> |--|--------|-------|-------|
> | CoOp  | 65.63  | 84.00  | 87.53 |
> | LoCoOp| 61.27  | 83.13 | 88.17 |
>
>
>
> For the Flower dataset, the ID accuracy of LoCoOp is considerably lower than that of CoOp. This is because the information similar to flowers (e.g., grass) might be removed as backgrounds during training on ImageNet. On the other hand, the ID accuracy of LoCoOp is higher than that of CoOp for the Pets dataset. The background in ImageNet is similar to that of Pets. In addition, many of the images in Pets have similar backgrounds. Therefore, LoCoOp can remove unnecessary information (e.g., common backgrounds) to identify fine-grained kinds of dogs, so the ID accuracy is improved.
>
> > Q3. Training time
>
> A3. The training efficiency is also one of the strengths of our LoCoOp.
>
> For a 1-shot setting on ImageNet-1K, LoCoOp takes about 13 minutes with a single A100 GPU.  For a 16-shot setting on ImageNet-1K, LoCoOp takes about 3.5 hours with a single A100 GPU.
>
> Therefore, LoCoOp is easy to implement in environments with limited GPU resources.

---

> > ### Comment · Reviewer_uDwi · 2023-08-13
> > **Reply to the authors' rebuttal**
> >
> > Thanks to the author for answering my doubts in detail. However, since the author only provided very few visualizations, I still have concerns about the accuracy of the regional positioning, so I decided to maintain the current score.

---

### Official Review · Reviewer_Lcii · 2023-07-07

**Soundness:** 3 good
**Presentation:** 3 good
**Contribution:** 3 good
**Rating:** 6
**Confidence:** 5

**Summary:**

This paper proposes a vision-language prompt learning method named local regularized context optimization (LoCoOp) for few-shot out-of-distribution detection. The proposed LoCoOp method performs OOD regularization  that uses the portions of CLIP local features as OOD features during training. The experimental results show the effectiveness of the proposed LoCoOp method over zero-shot and fully supervised methods.

**Strengths:**

- The addressed problem of few-shot out-of-distribution detection is interesting and practical.
- The proposed CLIP based few-shot OOD detection method with local regularized context optimization is well motivated and achieves good results.

**Weaknesses:**

- The proposed local regularized context optimization method is not specific to the few-shot setting, and can actually also be applied to fully supervised OOD detection setting, where the ID-irrelevant nuisances could also be learned. Then will the proposed method also improve the results in fully supervised setting? And what are the results?
- Similar to the CLIP based segmentation task in [55], the ID-irrelevant regions are predicted based on CLIP prediction results. However, the segmentation results may not be good, will that be a key obstacle to the proposed method?
- The training process of the proposed method is a little bit confusing. Does the L_coop loss apply to the whole image or only the ID-relevant regions?
- In Table 1, it shows that the performance of one-shot LoCoOp method is even worse than the zero-shot CL-MCM method (and only comparable for 16-shot LoCoOp) on iNaturalist dataset. What is the reason?

**Questions:**

- Will the proposed method also improve the results in fully supervised setting? And what are the results?
- Will the region segmentation results be a key obstacle to the proposed method?
- What is the reason that the performance of one-shot LoCoOp method is even worse than the zero-shot CL-MCM method (and only comparable for 16-shot LoCoOp) on iNaturalist dataset?

**Limitations:**

The authors have addressed the limitations.

---

> ### Author Rebuttal · Authors · 2023-08-08
>
> We thank the reviewer for the valuable comments and respond to them as follows. We will add suggested experiments and explanations in the updated manuscript.
>
> > Q1.  The effectiveness of LoCoOp in a fully supervised setting.
>
>
>
> A1. As the reviewer pointed out, our LoCoOp can be applied to a fully supervised setting (i.e., using all training data). However, Figure 2 shows that the improvement in performance will become slower after 16-shot training. This is due to the fact that even if many images are used for training, CoOp and LoCoOp only train text prompts that have only a few parameters, and a small number of samples is sufficient to reach the upper score.
>
> > Q2. Is the segmentation result a key obstacle to the proposed method?
>
>
>
> A2.  That is correct, and the accuracy of the segmentation results is a major factor.
>
> In our setting, unlike typical segmentation tasks, correctly guessing the segmentation label is not necessary. Hence, to account for potential errors in the segmentation results, we introduce Rank in Eq.(4) to allow segmentation errors up to top-K.
>
> We also attach some visualization results in the global response to show how well the OOD regions are extracted. This shows that our method can correctly extract OOD regions.
>
> > Q3. Does the L_coop loss apply to the whole image or only the ID-relevant regions?
>
>
>
> A3. Thanks for pointing this out. L_coop is the loss applied for the entire image.  We will add an explanation in the final version.
>
> > Q4. The reason for low performance on the iNaturalist dataset
>
> A4. The reason for this may be difficult to analyze. In the case of ResNet in Table 4, the detection performance of LoCoOp with GL-MCM is superior to that of zero-shot GL-MCM on iNaturalist, so it cannot be said that LoCoOp's training strategy is the main cause of the problem.
>
> Besides, comparisons of different backbone networks for OOD detection often have been explored [1, 2] but no conclusions have yet been drawn because of the different performance on each dataset.
>
> [1] Fort et al., Exploring the limits of out-of-distribution detection, NeurIPS 2021.
> [2] Hendrycks et al., Scaling out-of-distribution detection for real-world settings, ICML 2022.

---

> > ### Comment · Reviewer_Lcii · 2023-08-17
> > **Post rebuttal**
> >
> > The authors addressed most my concerns. So I keep my original rating and lean towards acceptance.

---

### Author Rebuttal · Authors · 2023-08-04

We would like to express our gratitude to the reviewers for giving excellent and positive comments and recognizing our contributions: “Few-shot out-of-distribution detection is interesting and practical” (**Lcii, uDwi, RaSR**), “LoCoOp is well motivated and interesting” (**Lcii, uDwi, wEUW, RaSR, epH7**), and “The effectiveness of LoCoOp is clear” (**Lcii, uDwi, RaSR, epH7**), and “This paper is well written” (**wEUW, RaSR, epH7**).

We write responses to each reviewer in each thread. Visualization is a common question, and we share the visualization results here.
## Visualization of OOD regions

Thanks for the really important point. We have attached the visualization results of OOD regions LoCoOp extracts in the pdf.  This shows that our method can correctly extract OOD regions.

As reviewers (**Lcii, uDwi, wEUW, RaSR**) pointed out, the performance of segmentation results is a key to our method. However, unlike normal segmentation tasks, it is not necessary to correctly guess the segmentation label in our setting. Therefore, to compensate for some errors in segmentation results, we introduce Rank in Eq. (4) in the main, which allows segmentation mistakes up to top-K.

We will add this figure and this explanation in the final version.

---

### Decision · Program_Chairs · 2023-09-21

**Decision:**

Accept (poster)

**Comment:**

The paper has received unanimous acceptance recommendations, with one accept, one weak accept, and three borderline accept ratings. The topic of few-shot out-of-distribution detection is considered highly relevant and interesting to the research community. Reviewers have lauded the simplicity and effectiveness of the method, which is substantiated by compelling experimental results.

The Area Chair concurs with the reviewers' assessments and joins them in recommending the acceptance of the paper.